# An Octopus-Derived Peptide with Antidiuretic Activity in Rats

**DOI:** 10.3390/md20050328

**Published:** 2022-05-17

**Authors:** Ye-Ji Kim, Jei Ha Lee, Seung-Hyun Jung, Ki Hyun Kim, Chang-Hoon Choi, Seonmi Jo, Dong Ho Woo

**Affiliations:** 1Research Center for Convergence Toxicology, Korea Institute of Toxicology, Daejeon 34114, Korea; yeji.kim@kitox.re.kr (Y.-J.K.); cchcom@kitox.re.kr (C.-H.C.); 2Human and Environmental Toxicology, University of Science and Technology, Daejeon 34114, Korea; 3Department of Genetic Resources, National Marine Biodiversity Institute of Korea, Seocheon 33662, Korea; jeiha@mabik.re.kr (J.H.L.); zebrajung@mabik.re.kr (S.-H.J.); tjckkh@naver.com (K.H.K.)

**Keywords:** cephalotocin, octopressin, antidiuretic, octopus, vasopressin

## Abstract

Discovering new drug candidates with high efficacy and few side effects is a major challenge in new drug development. The two evolutionarily related peptides oxytocin (OXT) and arginine vasopressin (AVP) are known to be associated with a variety of physiological and psychological processes via the association of OXT with three types of AVP receptors. Over decades, many synthetic analogs of these peptides have been designed and tested for therapeutic applications; however, only a few studies of their natural analogs have been performed. In this study, we investigated the bioactivity and usefulness of two natural OXT/AVP analogs that originate from the marine invertebrate *Octopus vulgaris*, named octopressin (OTP) and cephalotocin (CPT). By measuring the intracellular Ca^2+^ or cyclic AMP increase in each OXT/AVP receptor subtype–overexpressing cell, we found that CPT, but not OTP, acts as a selective agonist of human AVP type 1b and 2 receptors. This behavior is reminiscent of desmopressin, the most widely prescribed antidiuretic drug in the world. Similar to the case for desmopressin, a single intravenous tail injection of CPT into Sprague-Dawley rats reduced urine output and increased urinary osmolality. In conclusion, we suggest that CPT has a significant antidiuretic effect and that CPT might be beneficial for treating urological conditions such as nocturia, enuresis, and diabetes insipidus.

## 1. Introduction

Oxytocin (OXT) and arginine vasopressin (AVP) are nonapeptides that are well known as a uterine-contracting hormone and an antidiuretic hormone, respectively. The genes encoding these two peptides are usually located close to each other on the same chromosome and are thought to have arisen through the duplication of an ancestral gene, named vasotocin [1]. Consequently, the two active nonapeptides are very similar in structure in terms of internal disulfide bonds and amidated C-termini [2,3]; only two amino acids, at positions 3 and 8, distinguish OXT and AVP. Once secreted from the posterior pituitary into the bloodstream, these peptide hormones exert their actions via one oxytocin receptor (OXTR) and three vasopressin receptors (V1aR, V1bR, and V2R), which are differentially expressed in various tissues [4]. In addition to their emerging roles in sociality and emotion [5], these hormones have physiological consequences, including uterine contraction and breast milk ejection via OXTR [6], vasoconstriction and platelet aggregation via V1aR (also called V1R), pituitary adrenocorticotropic hormone secretion in response to stress via V1bR (also called V3R), and antidiuretic water reabsorption in the kidney via V2R [7]. Inappropriate receptor activation leads to diseases and medical conditions associated with the various physiological processes described above [6,7].

Due to the multi-receptor action of AVP, selective modulation of each OXT/AVP receptor subtype is usually considered a more effective therapy than administration of OXT or AVP itself. For example, desmopressin (1-deamino-8-D-arginine vasopressin, DDAVP), a synthetic analog of AVP that is a selective agonist for V1bR and V2R [8], has been widely used in the treatment of diabetes insipidus, enuresis (bed wetting), nocturia, von Willebrand disease, and hemophilia A [9,10]. Continuous efforts are still being made to develop new drugs with higher specificity and fewer side effects; thousands of OXT/AVP analogs have been synthesized and characterized for therapeutic applications [11]. In addition to synthetic analogs, various natural analogs have been found in divergent species and studied to elucidate their original biological roles [12,13]. Their chemical structures and encoding genes are well conserved throughout the animal kingdom, with little variation in amino acid sequence recorded [14]. It is plausible that these natural analogs can bind to any of the OXT/AVP receptor(s) and further promote or inhibit receptor activation [15]. If so, their use may vary depending on receptor selectivity. Indeed, conopressin-T from the tulip cone snail (*Conus tulipa*) and inotocin from the black garden ant (*Lasius niger*) were found to be selective antagonists of V1aR [16,17]. Better analogs were designed by using these peptides as templates and have been suggested as preclinical drug candidates [16,17]. However, aside from these cases, there have been very few studies on the usefulness of natural OXT/AVP analogs.

While most invertebrates carry only one gene encoding OXT/AVP superfamily peptides, the common octopus (*Octopus vulgaris*), a molluscan cephalopod with vertebrate-like organ systems and intelligent behaviors, is the first reported invertebrate to have two such genes [18]. These genes produce active nonapeptides, which are named octopressin (OTP) and cephalotocin (CPT) [18,19]. In the same previous study, OTP induced contractions of in vitro preparations of peripheral tissues, such as oviduct, aorta, and rectum [18]. OTP was also reported to evoke hyperactivity in chromatophore cells, rapid respiration, and jetting water from the siphon in the same species [20]. In another octopus species, *O. ocellatus* (a synonym of *Amphioctopus fangsiao*), OTP is relevant to osmoregulation as it decreases hemolymph osmolality [21]. CPT did not show any visible effect in two other octopus species; however, another cephalopod study showed that both OTP and CPT affect long-term memory in the common cuttlefish (*Sepia officinalis*) [22].

Although the effects of OTP and CPT on a few cephalopod species have been studied, as described above, whether these peptides can also be functional in other animal taxa, especially in mammals, has not been studied. Similar to other OXT/AVP analogs, these octopus peptides might have therapeutic potential by targeting the OXT/AVP system and thereby modulating physiological functions. However, this possibility has not been investigated thus far. To address this issue, we examined the agonistic and antagonistic effects of OTP and CPT on each type of human OXT/AVP receptor in vitro. Based on the results, CPT was administered to Sprague-Dawley (SD) rats to test its efficacy in vivo. Finally, the viability of cells cultured with CPT was also measured to test for any cytotoxic effects.

## 2. Results

### 2.1. OTP Does Not Affect Human OXT/AVP Receptors

The octopus peptides OTP and CPT are natural analogs of human OXT/AVP. All of them consist of nine amino acids and have a disulfide bond between cysteine residues 1 and 6. The glycine at the C-terminus is amidated. Considering that other natural analogs, such as conopressin-T and inotocin, are selective antagonists of V1aR [16,17], we first performed an in vitro functional assay to determine whether OTP acts on any of the human OXT/AVP receptors. Cultured mammalian cells expressing each human OXT/AVP receptor were treated with various concentrations of high-purity OTP, as described in the Materials and Methods. We found that OTP had no or minimal effect on human OXT/AVP receptors (Figure 1A–D). Up to 50 μM, OTP did not seem to induce any OXTR-, V1aR-, or V1bR-mediated intracellular Ca^2+^ increases (Figure 1A–C, black circles). Only V2R-mediated cAMP accumulation was induced at 50 μM OTP, but the response was small (<30%, Figure 1D, black circles). In addition, OTP could not inhibit the stimulant-induced activation of V1aR, V1bR, or V2R at any of the concentrations tested (Figure 1A–C, white circles). Only OXTR activation was inhibited at 50 μM OTP, but the inhibition rate was low (approximately 30%, Figure 1D, white circles). The effects of OTP were not sufficient to conclude that it had an agonist or antagonist effect on human OXT/AVP receptors.

### 2.2. CPT Selectively Activates Human V1bR and V2R

Next, we performed the same functional assay as above by using various concentrations of high-purity CPT. CPT clearly showed dose-dependent effects on each human OXT/AVP receptor (Figure 2A–D). CPT activated human V1bR with an EC_50_ of 10.02 nM (Figure 2C, black circles). CPT also activated human V2R with an EC_50_ of 12.09 nM (Figure 2D, black circles). Interestingly, OXTR and V1aR were not activated by CPT at concentrations up to 50 μM (Figure 2A,B, black circles). We found that rather than acting as an agonist for these two receptors, CPT had a weak antagonist effect against OXTR (Figure 2A, white circles) and against V1aR (Figure 2B, white circles). The IC_50_ values were more than 50 μM for OXTR and approximately 1.23 μM for V1aR. Although CPT is antagonistic to V1aR activity, similar to other natural analogs in previous studies [16,17], the much lower EC_50_ for V1bR and V2R indicates that the main target of CPT would be V1bR and V2R rather than OXTR and V1aR. Indeed, when we calculated whether CPT at a certain concentration was able to activate or inhibit each receptor, a nanomolar level of CPT was expected to selectively activate V1bR and V2R (~80%) without notable effects on OXTR and V1aR (<20%) (Figure 2E). At the micromolar level, full activation of V1bR and V2R, but notable inhibition of V1aR activity (≥50%) by CPT was also expected (Figure 2F). If applied at a very high concentration (≥100 μM), full activation of V1bR and V2R and significant inhibition of OXTR and V1aR were expected (Figure 2G). Therefore, we conclude that only CPT, and not OTP, may effectively and selectively regulate the function of human OXT/AVP receptors and can be considered a candidate bioactive drug targeting V1bR and V2R.

### 2.3. A Single Tail Intravenous Injection of CPT Reduced the Collected Urine Output in SD Rats

V1aR is distributed in vascular and hepatic regions. V1bR is expressed in the anterior pituitary in the brain. V2R is expressed mainly in the kidney [23]. The agonist effect of CPT, which is selective for human V1bR and human V2R, is reminiscent of the effect of desmopressin, the most widely prescribed antidiuretic drug in the world. Desmopressin is well known as a selective agonist for human V1bR (EC_50_ = 11.4 nM) and human V2R (EC_50_ = 23.9 nM) [8]. It exerts its antidiuretic effect mainly through the V2R expressed in the kidney. Therefore, we wondered if CPT would have a similar antidiuretic effect to desmopressin. To test this possibility, we intravenously injected CPT into SD rats and measured the volume and osmolality of collected urine outputs. Prior to administration, SD rats were fasted for 20–22 h to maximize the efficacy of the CPT. CPT (1 mg/mL/kg) was administered once using the tail microvein (Figure 3A). Micro intravenous tail administration was performed by trained researchers (IV injection in Figure 3A,B). Rats administered CPT were left in metabolic cages with drinking water only for 20–24 h. The metabolic cage can collect urine only for one day (Figure 3C). A single injection of CPT significantly reduced urine volume and increased solute concentration (Figure 3D,F). Over 24 h, the CPT group had an average urine volume of 10.79 mL, nearly 49% lower than the control group’s mean urine volume of 21.14 mL (Figure 3D, ** *p* < 0.01). Three identical independent experiments showed similar reductions in urine volume (Figure 3E, * *p* < 0.05). Conversely, the average urine osmolality of the CPT group was 946 mOsm/kg, a 43% increase compared to the control group average osmolality of 536 mOsm/kg (Figure 3f, * *p* < 0.05). Three identical independent experiments showed similar changes in osmolyte concentration (Figure 3G, * *p* < 0.05, ** *p* < 0.01). In conclusion, CPT has a significant antidiuretic effect in vivo and might have therapeutic potential for the treatment of medical conditions related to excessive urination. To optimize the CPT effect, the rat’s weight must be 200 g or greater (Appendix A, ** *p* < 0.01 *** *p* < 0.001), the floor of the housing must be polycarbonate (Appendix A, * *p* < 0.05 ** *p* < 0.01), and the fasting duration must be more than 17 h (Appendix A, * *p* < 0.05 **** *p* < 0.0001). Micro intravenous tail administration of 0.001 mg/mL/kg AVP also shows an antidiuretic effect but is occasionally lethal. Therefore, desmopressin was used as a positive control for antidiuretic effects. From these results, it can be inferred that AVP activated both vascular V1AR and kidney V2R, and desmopressin activated kidney V2R. Interestingly, CPT showed a mild antidiuretic effect compared to DP (Figure 3D,F). A single intravenous injection of CPT did not kill the experimental animals.

### 2.4. CPT Does Not Alter the Viability of HEK293T Human Kidney Cells

To test whether the antidiuretic effect of CPT was due to toxicity against V2R-expressing kidney cells, we measured the cell viability of human kidney HEK293T cells after CPT treatment. Several concentrations of CPT were tested, and the maximal concentration was the same as that in the receptor function assay in Figure 1 and Figure 2. After 24, 48, and 72 h of CPT treatment, the cell morphology seemed to be unchanged on imaging (Figure 4A,B). By performing a CCK-8 assay, we found that cell viability was not significantly affected by CPT treatment (Figure 4C). Over 98% of HEK293T cells were viable at 24 and 48 h after treatment with AVP or CPT, which was not significantly different from the viability of untreated controls. After 72 h, AVP or CPT treatment at 0.1–10 μM tended to slightly increase the viability when compared to the negative control treatment (PBS). However, there were still no significant differences between AVP and CPT at any concentration tested (Figure 4C). These results indicate that the antidiuretic effect of CPT was likely expressed within a range that did not specifically harm the kidney cells. Moreover, the absence of cellular toxicity up to 50 μM CPT suggests that CPT could be a candidate alternative antidiuretic agent.

## 3. Discussion

Currently, desmopressin is the antidiuretic drug of choice for the treatment of excessive and/or uncontrolled urination. It is an artificial analog of AVP designed by deamination of the first amino acid and replacement of the last L-arginine with D-arginine [24]. Desmopressin was carefully chosen after screening many analogs for activity, side effects, and effect duration [25]. Its prolonged half-life and low V1aR-mediated pressor activity have enabled desmopressin to be widely used for the last 45 years. Although desmopressin is generally considered safe and efficacious in the treatment of nocturia, enuresis, and diabetes insipidus, hyponatremia is known to be a serious adverse effect [26]. Careful instructions about fluid intake should be given to minimize the incidence of hyponatremia and other clinical symptoms of water intoxication, such as vomiting, headache, and decreased consciousness [27]. It has been suggested that strengthening of the bioactivity of desmopressin, either by prolonged half-life or overdose exposure, may increase the risk of water intoxication [26,27]. For example, the pharmacokinetics of desmopressin may be altered by aging or underlying disease, leading to drug overexposure [28,29,30]. This means that the development of alternative antidiuretic drugs is still needed.

As a potential alternative drug candidate, CPT showed a milder antidiuretic effect than desmopressin in this study (Figure 3). However, there is a limit to directly extrapolating the results of this experiment in rats to efficiency in humans: V2R is located in the kidney and is well known to mediate antidiuretic action by increasing the membrane trafficking of aquaporin-2 (AQP2) water channels and by increasing the protein abundance of AQP2 [31,32]. V1bR, the other target of CPT and desmopressin, is also functional in the kidney, although the expression level is very low [33]. Administration of a selective V1bR agonist results in a diuretic effect [33]. Therefore, the balance of V1bR activity and V2R activity should be considered when discussing renal water regulation. For reference, EC_50_ of AVP on human V1bR and V2R is 1.51 nM and 2.87 nM and that of desmopressin on human V1bR and V2R is 11.4 nM and 23.9 nM, respectively [8]. In this study, EC_50_ of CPT was found to be 10.02 nM on human V1bR and 12.09 nM on human V2R, indicating similar potency to desmopressin (Figure 2). EC_50_ of AVP is 1.31 nM on rat V1bR and 43.5 nM on rat V2R [8]. Desmopressin has a 4-fold greater EC_50_ for rat V1bR than for human V1bR [8]. Likewise, CPT may have a different (greater or lesser) EC_50_ and binding affinity for rat V1bR than for human V1bR. Additional study is needed to explain how CPT interacts with rat V1bR and to investigate its antidiuretic efficiency in humans. 

When administered intravenously, the half-life of AVP is generally known to be several minutes. In the case of desmopressin, the deamination of the N-terminus and a presence of D-form amino acid greatly extends its half-life to several hours. By comparison, we could not find any indications that CPT would have a significantly different half-life compared to AVP. CPT has the same chemical structure as AVP except for only two out of nine amino acids. According to in silico analysis of amino acid sequences using online peptide analyzing tools (provided by Thermo Fisher Scientific, Waltham, MA, USA and GenScript, Piscataway, NJ, USA), there is no remarkable difference between CPT and AVP in terms of hydrophobicity, molecular weight, isoelectric point, net charge at pH7, and the content of less stable amino acids. These suggest CPT will have a similar level of stability as AVP and a natural, short half-life like AVP. A short half-life is often problematic for drug development, but in this case, it is not critical because the antidiuretic effect is not directly mediated by V2R itself but by the V2R-driven increase in AQP2 [28], as mentioned above. The antidiuretic effect of CPT may persist as long as AQP2 is functional, even after V2R activity has stopped. In fact, turning off the V2R activity after AQP2 has been expressed would be beneficial in minimizing possible side effects and safety concerns. In addition, unlike in the past, when artificial analogs had to be synthesized to overcome the limitations of natural peptides, advances in peptide drug delivery systems can extend a short half-life or resolve other problems, such as low oral bioavailability [34,35]. Consequently, we suggest that CPT could be developed as an alternative or adjuvant antidiuretic agent.

To date, the roles of the OXT/AVP superfamily peptides of nonhuman animals have mostly been explored in an evolutionary context. For example, annetocin induces spawning-like behavior when injected into the annelid earthworm [12], and asterotocin induces feeding behavior in an echinoderm starfish [13]. However, with the exception of a few V1aR-antagonist peptides, such as conopressin-T [16] and inotocin [17], as mentioned previously, it seems that few peptides from invertebrates have been considered in attempts to target human OXT/AVP receptors. Porcine lypressin is the only example of a natural peptide available for treating diabetes insipidus [36]. Here, we focused on a peptide naturally occurring in the octopus. This octopus CPT is the first example of a V1bR- and V2R-selective natural antidiuretic substance. Since the diversity of these superfamily peptides is likely highly consistent with biodiversity, it is worth seeking more antidiuretic candidates in various organisms. We hope that this study facilitates this field of research, connecting biodiversity to drug discovery.

## 4. Materials and Methods

### 4.1. Peptides

As described in Table 1, high purity of OXT, AVP, OTP, and CPT were synthesized by AnyGen (Gwangju, Korea) using solid-phase peptide synthesis. The purity and molecular masses of the peptides were determined using high-performance liquid chromatography (HPLC) and matrix-assisted laser desorption ionization time-of-flight mass spectrometry (MALDI-TOF) (Shimadzu, Kyoto, Japan). Desmopressin was obtained from Tocris Bioscience (Bristol, UK). The freeze-dried peptide powders were kept frozen and dissolved in DMSO (for in vitro receptor functional assays) or pure water (for in vivo experiments and for cell viability assays) before use. The purity of the synthetic peptides was proved to be over 98%. The purity of desmopressin was more than 95% according to the manufacturer’s instructions.

### 4.2. In Vitro Assay

#### 4.2.1. In Vitro OXT/AVP Receptor Functional Assay

Agonist or antagonist effects of OTP and CPT on human OXT/AVP receptors were evaluated at Eurofins Cerep (Celle L’Evescault, France). Peptide concentrations for treatment were determined on the basis of the known 50% effective concentrations (EC_50_) of OXT and AVP. Detailed information about the cells tested and other experimental conditions are described in Table 2. Briefly, cells were transfected with a recombinant expression vector capable of expressing each type of human OXT/AVP receptor. To evaluate the agonist effects on OXTR, V1aR, and V1bR, the intracellular Ca^2+^ increase after treatment with OTP or CPT was measured by fluorimetry. To evaluate the antagonist effect on these same receptors, an intracellular Ca^2+^ increase was artificially induced by the stimulant (OXT or AVP), and then the Ca^2+^ decrease stimulated by cotreatment with OTP or CPT was measured by fluorimetry. For V2R receptors, cells were treated with peptides with or without the stimulant for 30 min. After that, cyclic AMP (cAMP) concentrations were measured by homogeneous time-resolved fluorescence (HTRF). The results were represented as the percent submaximal response achieved by the stimulant. All experiments were performed in triplicate. Data are presented as the mean ± standard error of the mean. The 50% effective concentration (EC_50_) and 50% inhibitory concentration (IC_50_) values were determined using a 4-parameter logistic fitting of a dose–response curve using SigmaPlot 10.0 (Systat Software, Palo Alto, CA, USA).

#### 4.2.2. CCK-8 Cell Viability Assay

Human kidney HEK293T cells were obtained from the American Type Culture Collection (ATCC, Manassas, VA, USA). The 293T cells were grown in DMEM supplemented with 10% fetal bovine serum (HyClone, Logan, UT, USA) and penicillin–streptomycin. Cells were incubated at 37 °C in a humidified atmosphere with 95% air/5% CO_2_. HEK293T cells (5 × 10^3^) were incubated in 96-well plates to measure cell viability using a Cell Counting Kit-8 (CCK-8; Dojindo Laboratories, Kumamoto, Japan). After overnight incubation, the medium was changed to fresh medium containing CPT at doses of 50~0.1 μM in three wells each. Then, the cells were incubated for 24, 48, or 72 h and photographed under a phase-contrast microscope. We added 100 μL of fresh medium and 10 μL of CCK-8 solution to each well. After incubation for 3 h at 37 °C, the optical density (OD) at 450 nm was measured using a Spectramax i3x instrument (Molecular Devices, San Jose, CA, USA). The cell viability was calculated as follows: cell viability (%) = (OD (experiment) − OD (blank))/(OD (control) − OD (blank)) × 100%

### 4.3. Antidiuretic Experiments

Five-week-old male Sprague-Dawley (SD) rats were acclimatized for 1–2 weeks after acquisition until reaching 200 g. The SD rats were then grouped based on body weight. To maximize the peptide effect on antidiuretic activity, SD rats were fasted for 12 h. A highly trained researcher administered 1 mg/mL of OTP or CPT or 0.01 mg/mL of desmopressin dissolved in saline solution through tail intravenous injection using a 26 gauge syringe. The rats were housed in polycarbonate cages with wood shaving bedding. Food pellets and water were provided ad libitum. The rats were placed into metabolic cages (KyeRyong Science, Daejeon, Korea) for 22–24 h to collect urine for measurement of the daily urine output and osmolality (Micro-Osmometer, Model 210, Advanced Instruments, Inc., Norwood, MA, USA). All experiments were carried out in accordance with current international guidelines for the care and use of experimental animals, and the experimental protocols were approved by the Institutional Animal Care and Use Committee of KIT (KIT IACUC reference numbers 1901-0028).

### 4.4. Data Analysis

Experimental data are presented as the mean, and the error bars represent the experimental standard error of the mean; both values were calculated using Excel 2016 software (Microsoft, Redmond, WA, USA). The EC_50_ (concentration producing a half-maximal response) and IC_50_ (concentration causing a half-maximal inhibition of the control agonist response) values were determined by 4-parameter logistic fitting of the dose–response curve using SigmaPlot 10.0 (Systat Software, Palo Alto, CA, USA). GraphPad Prism version 8 was used for *t* tests and one-way ANOVA post hoc tests. Comparisons were performed using two-tailed paired Student’s *t* tests or one-way ANOVA. *p*  <  0.05 was considered significant: * *p*  <  0.05; ** *p*  <  0.01; *** *p*  <  0.001, **** *p* < 0.0001 as indicated in individual figures.

## 5. Conclusions

In this study, we found that CPT is a selective agonist for human V1bR and V2R at low nanomolar concentrations (Figure 1 and Figure 2), similar to the widely prescribed antidiuretic drug desmopressin. In the rats we tested, a single intravenous injection of CPT showed a significant antidiuretic effect (Figure 3). While AVP injection occasionally caused death in rats (data not shown), the same doses of desmopressin and CPT did not cause any death or signs of toxicity. In addition, cell viability was maintained in CPT-treated human kidney HEK293T cells (Figure 4). Therefore, we expect that CPT can be considered a novel candidate antidiuretic agent.

## 6. Patents

Y. Kim, C. H. Choi, S. H. Jung, K. H. Kim, S. Jo and D. H. Woo are listed as the inventors on patent application PCT/KR2021/004415 and Korean patent 10-2202346, which cover “Compositions for preventing, improving or treating urination-related disease comprising cephalotocin”.

## Figures and Tables

**Figure 1 marinedrugs-20-00328-f001:**
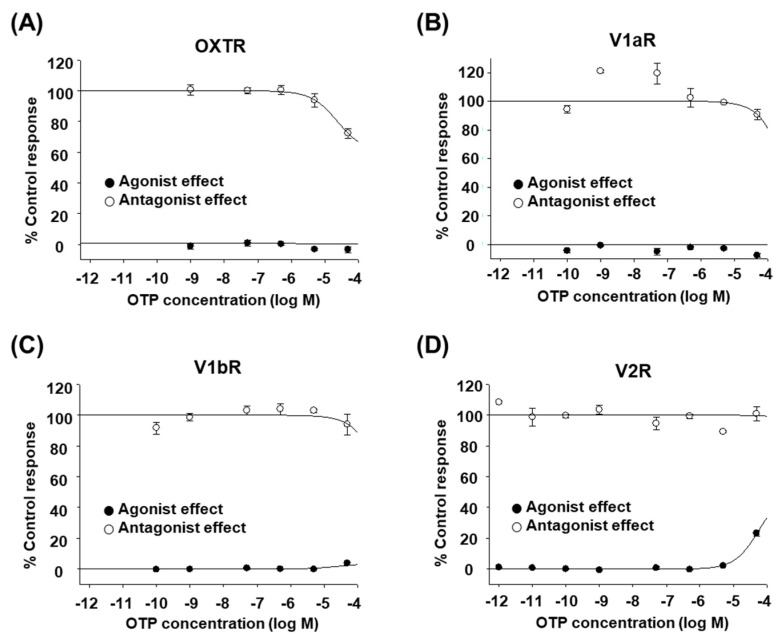
Effect of OTP on human OXT/AVP receptors. (**A**–**D**) in vitro functional assay of human OXT/AVP receptors stimulated by different OTP concentrations. The agonist effect (●) and antagonist effect (○) of OTP on human OXT/AVP receptors are presented as the percentage of the control response. At each OTP concentration, (**A**) changes in OXTR-mediated intracellular Ca^2+^ release, (**B**) V1aR-mediated intracellular Ca^2+^ release, (**C**) V1bR-mediated intracellular Ca^2+^ release, and (**D**) V2R-mediated cAMP accumulation were measured. The data shown are the mean ± SEM of triplicate experiments.

**Figure 2 marinedrugs-20-00328-f002:**
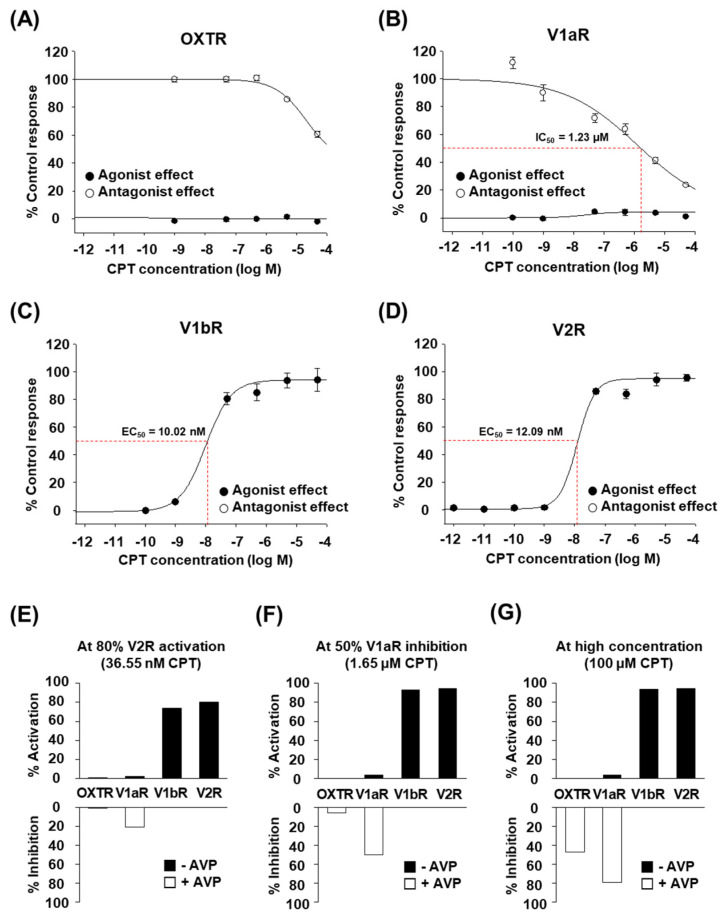
Effect of CPT on human OXT/AVP receptors. (**A**–**D**) in vitro functional assay of human OXT/AVP receptors stimulated by different CPT concentrations. Agonist effect (●) and antagonist effect (○) of CPT on human OXT/AVP receptors, represented by the percentage of the control response. At each CPT concentration, (**A**) changes in OXTR-mediated intracellular Ca^2+^ release, (**B**) V1aR-mediated intracellular Ca^2+^ release, (**C**) V1bR-mediated intracellular Ca^2+^ release, and (**D**) V2R-mediated cAMP accumulation were measured. Data are the mean ± SEM of triplicate experiments. (**E**–**G**) Degree of activation or inhibition of each OXT/AVP receptor subtype calculated by curve-fitting of the experimental data above. (**E**) At the CPT concentration (36.55 nM) required for 80% V2R activation. (**F**) At the CPT concentration (1.65 μM) required for 50% V1aR inhibition. (**G**) At 100 μM CPT.

**Figure 3 marinedrugs-20-00328-f003:**
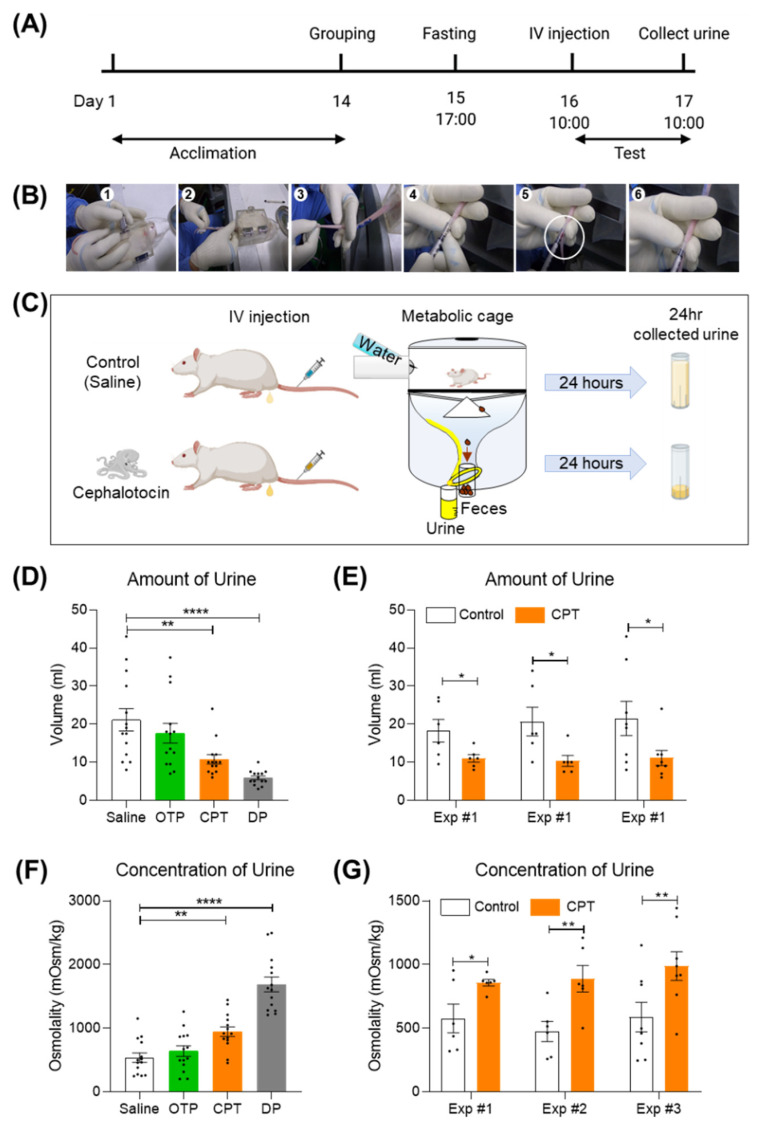
A single tail intravenous injection of CPT reduced daily urine output in SD rats. (**A**) Experimental schedule, IV injection, sequential images for the procedure of finding microvessels in the tail for injecting CPT. (**B**) Description of the experimental method for finding a microvein for injection. (**C**) A single tail intravenous injection of CPT reduced the volume of urine collected in one day. (**D**) Summary bar graph for the amount of urine collected (one-way ANOVA F (3, 52) = 10.96, *p* < 0.0001 followed by Dunnett’s multiple comparisons test ** *p* = 0.0024 saline & CPT, **** *p* < 0.0001 saline & DP). Black circles represent each individual value. (**E**) Summary bar graph for independent experiments to measure the amount of urine (unpaired *t* test * *p* = 0.0436 for exp#1, * *p* = 0.0287 for exp#2, * *p* = 0.0183 for exp#3). (**F**) Summary bar graph for a one-day collection of osmolytes (one-way ANOVA F (3, 52) = 33.66, *p* < 0.0001 followed by Dunnett’s multiple comparisons test ** *p* = 0.0063 saline & CPT, **** *p* < 0.0001 saline & DP). (**G**) Summary bar graph for independent experiments for the amount of urine (unpaired *t* test * *p* = 0.0355 for exp#1, ** *p* = 0.0097 for exp#2, ** *p* = 0.0024 for exp#3). Data are the mean ± SEM. CPT, cephalotocin; OTP, octopressin; DP, desmopressin.

**Figure 4 marinedrugs-20-00328-f004:**
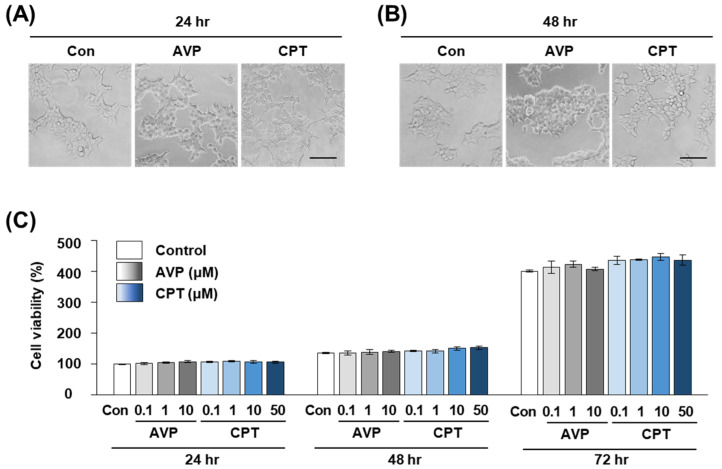
Viability of HEK293T cells treated with AVP or CPT. (**A**,**B**) Cellular morphology of HEK293T cells at (**A**) 24 and (**B**) 48 h after treatment with 10 μM AVP or CPT. Con: PBS as a negative control. Scale bar = 100 μm. (**C**) Cell viability measured by CCK-8 assay at 24, 48 or 72 h after treatment with different concentrations of AVP or CPT. Con: PBS as a negative control. Data are the mean ± SEM of triplicate experiments.

**Table 1 marinedrugs-20-00328-t001:** OXT/AVP and their analogs used in this study.

Name	Sequence	Purity	Species
Oxytocin (OXT)	CYIQNCPLG-NH_2_ (disulfide bond)	>98%	Mammals
Vasopressin (AVP)	CYFQNCPRG-NH_2_ (disulfide bond)	>98%	Mammals
Desmopressin (DP)	Mpr-YFQNCP(D-Arg)G-NH_2_ (disulfide bond)	>95%	Synthetic (From AVP, deamination of 1-C and substitution of 8-D-R)
Cephalotocin (CTP)	CYFRNCPIG-NH_2_ (disulfide bond)	>98%	*O. vulgaris*
Octopressin (OTP)	CYWTSCPIG-NH_2_ (disulfide bond)	>98%	*O. vulgaris*

**Table 2 marinedrugs-20-00328-t002:** Experimental conditions for the OXT/AVP receptor functional assay.

Receptors	Cell Lines	Stimulus (Full Activation)	Incubation	Measured Component	Detection Method	Ref
OXTR (h) (agonist effect)	ECV304	none (1 µM OXT)	RT	intracellular [Ca^2+^]	Fluorimetry	[37]
OXTR (h) (antagonist effect)	ECV304	30 nM OXT	RT	intracellular [Ca^2+^]	Fluorimetry	
V1aR (h) (agonist effect)	CHO	none (1 µM AVP)	RT	intracellular [Ca^2+^]	Fluorimetry	[38]
V1aR (h) (antagonist effect)	CHO	10 nM AVP	RT	intracellular [Ca^2+^]	Fluorimetry	
V1bR (h) (agonist effect)	RBL	none (0.1 µM AVP)	RT	intracellular [Ca^2+^]	Fluorimetry	[39]
V1bR (h) (antagonist effect)	RBL	5 nM AVP	RT	intracellular [Ca^2+^]	Fluorimetry	
V2R (h) (agonist effect)	CHO	none (1 nM AVP)	30 min RT	cAMP	HTRF	[40]
V2R (h) (antagonist effect)	CHO	0.03 nM AVP	30 min RT	cAMP	HTRF	

## Data Availability

Not applicable.

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
