# Peer review of "An Octopus-Derived Peptide with Antidiuretic Activity in Rats"

_marinedrugs, 2022, doi:10.3390/md20050328_

Round 1

Reviewer 1 Report

The manuscript is devoted to the study of the biological activity of two peptides, natural analogues of oxytocin and vasopressin - octopressin and cephalotocin, isolated from marine invertebrate Octopus vulgaris. Experiments have been carried out in the in vitro model system on mammalian cells as well as in vivo in rats. It is shown that the CPT by activity and mechanism of action is similar to the vasopressin analogue Desmopressin.

Experiments are well planned, have appropriate controls. The manuscript is well written and has sufficient illustrative material and can be published in its current form.

One minor note: All synthesized peptides appear to be in the form of trifluoroacetate salts. Desmopressin, sold by Tocris Bioscience (CAS 16679-58-6), is supplied in the form of acetate salt. Since the basis of the work is the comparison of effective concentrations of the peptides, the question of the preparation of peptides solutions becomes important. I did not find a description in the text. Was there an experimental determination of the actual concentration of peptides in solutions after their preparation?

Author Response

à Please see blue color of correction as response to reviewer comments in main text.

1st reviewer

The manuscript is devoted to the study of the biological activity of two peptides, natural analogues of oxytocin and vasopressin - octopressin and cephalotocin, isolated from marine invertebrate Octopus vulgaris. Experiments have been carried out in the in vitro model system on mammalian cells as well as in vivo in rats. It is shown that the CPT by activity and mechanism of action is similar to the vasopressin analogue Desmopressin.

Experiments are well planned, have appropriate controls. The manuscript is well written and has sufficient illustrative material and can be published in its current form.

à We thank the reviewer for appreciating the completeness of our study.

One minor note: All synthesized peptides appear to be in the form of trifluoroacetate salts. Desmopressin, sold by Tocris Bioscience (CAS 16679-58-6), is supplied in the form of acetate salt. Since the basis of the work is the comparison of effective concentrations of the peptides, the question of the preparation of peptides solutions becomes important. I did not find a description in the text. Was there an experimental determination of the actual concentration of peptides in solutions after their preparation?

à We agree with that the preparation of peptide solutions is important. Being measured by HPLC (at manufacturer), purity of each peptide powder was shown in Table 1. Although we did not directly measure the final working concentrations again, the high purity of peptide powder above 95% or 98% (used in this study) could be considered almost pure. We can consider that the concentration of the actual peptide in the working solution would be lower by 2–5%, but it is unlikely that presence of the counter ion (TFA or acetate) in subtle concentration substantially affect the conclusion of this study. In response to the reviewer’s point, we add a description in the text of the revised manuscript as follows: “The purity of the synthetic peptides was proved to be over 98%. The purity of desmopressin was more than 95% according to the manufacturer's instructions.”

à In addition, to avoid confusion, we removed "Working conc" from Table 1 in the revised manuscript because the concentrations of peptides used in each experiment are redundantly described in the "Materials and Methods" section. We also corrected the typo that purity of desmopressin powder, from >98% to >95%.

Reviewer 2 Report

Minor

The experimental results are clean, and the manuscript is clearly written. But there are several corrections required, and a few such errors I caught (not concentrating and noting everything) are listed below as examples.

- Line 149, change “kidney [23]” to “kidney[23]”, which consistent with other parts of the manuscript.

- Line 329 and 336, the format of“℃”should be consistent.

- Line 331, (5×103) should be (5×103).

- The number of “EC50” and “IC50” in the manuscript need subscript.

- “Ca2+” in the manuscript should be “Ca2+”.

- The authors used “p” in line 164 and 166, but “P” in line 168.

- Line 277, “The freeze-dried peptide powders were kept frozen and dissolved in DMSO (for in vitro assays) or pure water (for in vivo experiments) before use”, why powders dissolved in DMSO for in vitro assays, but pure water for in vivo experiments?

- The authors should place 4.4 in the last section of Materials and Methods.

Major

The authors investigated the bioactivity and usefulness of OTP and CPT which originate from the marine invertebrate Octopus vulgaris. They found that CPT, but not OTP, acts as a selective agonist of human AVP type 1b and 2 receptors. The activity of CPT is strong and selective, and nontoxic to the kidney cells HEK293T, which allow it to be a candidate alternative antidiuretic agent. However, in the discussion, the assumption “As a natural peptide, it is expected that CPT will have a short half-life comparable to that of OXT or AVP” without any experimental evidence is not appropriate, especially, the author hope to find an alternative antidiuretic drugs for desmopressin with prolonged half-life. Thus, it should compare the stability among these peptides. in addition, it is also necessary to prove CPT interacts with V1bR and V2R and depict how CPT interacts with V1bR and V2R.

In the in-vitro OXT/AVP receptor functional assay, the authors used human oxytocin/vasopressin receptor instead of rat those, which can’t perfectly explain the mechanism of rat experiment results. Especially, the EC50 values of CPT against V1bR and V2R are very close with human oxytocin/vasopressin receptors in Figure 2C and D. That CPT may have a different (greater or lesser) EC50 and binding affinity for rat V1bR than for human V1bR need be verified.

Author Response

à Please see blue color of correction as response to reviewer comments in main text.

2nd reviewer

Minor

The experimental results are clean, and the manuscript is clearly written. But there are several corrections required, and a few such errors I caught (not concentrating and noting everything) are listed below as examples.

à We thank the reviewer for appreciating that our results and writings are clear. We corrected the errors pointed out by the reviewer and checked the entire manuscript again.

- Line 149, change “kidney [23]” to “kidney[23]”, which consistent with other parts of the manuscript.

à We removed a space after kidney.

- Line 329 and 336, the format of“℃”should be consistent.

à The Line must be 336, We removed a space before ℃.

- Line 331, (5×103) should be (5×103).

à The Line must be 330, superscript is corrected in text.

- The number of “EC50” and “IC50” in the manuscript need subscript.

à We corrected plain to subscripts as EC50 and IC50 in text.

- “Ca2+” in the manuscript should be “Ca2+”.

à Ca2+ was corrected to Ca2+.

- The authors used “p” in line 164 and 166, but “P” in line 168.

à We appreciated it. A p was corrected to P in line 164, 165.

- Line 277, “The freeze-dried peptide powders were kept frozen and dissolved in DMSO (for in vitro assays) or pure water (for in vivo experiments) before use”, why powders dissolved in DMSO for in vitro assays, but pure water for in vivo experiments?

à It was just a practical matter. The peptides used in this study were well soluble in water. So, we prepared the solution by freshly dissolving the peptide in pure water for in vivo experiments and cell viability assays. In case of in vitro receptor functional assays performed at Eurofins Cerep, making DMSO stock solution is the standard protocol. The stock solutions were diluted to the final DMSO concentration which does not affect the experimental results. In the revised manuscript, we modified the sentence to accurately describe in which experiment each solvent was used as follows: “The freeze-dried peptide powders were kept frozen and dissolved in DMSO (for in vitro receptor functional assays) or pure water (for in vivo experiments and for cell viability assays) before use.”

- The authors should place 4.4 in the last section of Materials and Methods.

à We rearranged materials and methods.

Major

The authors investigated the bioactivity and usefulness of OTP and CPT which originate from the marine invertebrate Octopus vulgaris. They found that CPT, but not OTP, acts as a selective agonist of human AVP type 1b and 2 receptors. The activity of CPT is strong and selective, and nontoxic to the kidney cells HEK293T, which allow it to be a candidate alternative antidiuretic agent. However, in the discussion, the assumption “As a natural peptide, it is expected that CPT will have a short half-life comparable to that of OXT or AVP” without any experimental evidence is not appropriate, especially, the author hope to find an alternative antidiuretic drugs for desmopressin with prolonged half-life. Thus, it should compare the stability among these peptides. in addition, it is also necessary to prove CPT interacts with V1bR and V2R and depict how CPT interacts with V1bR and V2R.

à We appreciate the constructive comment. In the revised manuscript, we modified the sentence as follows: "Because CPT has a structure similar to OXT/AVP and differs in that only a few natural amino acid portions are made of other natural amino acids, it is expected that CPT will have a similar stability and a natural, short half-life comparable to that of OXT or AVP." Elucidating the detailed mechanisms of how CPT interacts with V1bR and V2R is an interesting research topic. However, it is beyond the scope of this study, so we mentioned it in the Discussion section in the initial manuscript. We hope such follow-up studies and preclinical studies will be organized in the future.

In the in-vitro OXT/AVP receptor functional assay, the authors used human oxytocin/vasopressin receptor instead of rat those, which can’t perfectly explain the mechanism of rat experiment results. Especially, the EC50 values of CPT against V1bR and V2R are very close with human oxytocin/vasopressin receptors in Figure 2C and D. That CPT may have a different (greater or lesser) EC50 and binding affinity for rat V1bR than for human V1bR need be verified.

à We do agree with the reviewer’s point that human receptor experiments cannot perfectly explain the mechanism of rat experimental results. CPT may have a different (greater or lesser) EC50 and binding affinity for rat V1bR than for human V1bR. We already described about it in the Discussion section of the initial manuscript. Please consider that CPT is an antidiuretic candidate for “human use” and that the animal experiment is a very initial step testing those possibility. Verifying a different binding affinity to rat and human receptors is interesting and meaningful, but it is not critical at the present stage. We would like to emphasize that we found the potential for marine-derived CPT to be used as an antidiuretic drug. Optimization of dosage and delivery methods will be needed. Again, we thank the reviewer for the enthusiastic comment and suggestion.

Reviewer 3 Report

In the present study, authors investigated an octopus-derived peptide with antidiuretic activity in rats. The article has some questions as follows:

  1. In the abstract, “Octopus vulgaris” should be used in italics, and “Ca2+” should be used in superscripts.
  2. In the text, “Conus tulipa”, “Lasius niger”, “Octopus vulgaris”, “O. ocellatus”, “Amphioctopus fangsiao”, “Amphioctopus fangsiao” should be used in italics. And, “Ca2+”, “5×103” should be used in superscripts in the manuscript. “EC50”, “IC50” and “CO2” should be used in subscripts in the manuscript. “℃” should be used in its current form.
  3. “in vitro” and “in vivo” should be used in italics in the manuscript.
  4. Comparisons with similar previous studies should be made in the discussion section, such as EC50, IC50, and so on.

Author Response

à Please see blue color of correction as response to reviewer comments in main text.

3rd reviewer

In the present study, authors investigated an octopus-derived peptide with antidiuretic activity in rats. The article has some questions as follows:

  1. In the abstract, “Octopus vulgaris” should be used in italics, and “Ca2+” should be used in superscripts.

à We corrected Octopus vulgaris as italicized in abstract.

  1. In the text, “Conus tulipa”, “Lasius niger”, “Octopus vulgaris”, “O. ocellatus”, “Amphioctopus fangsiao”, “Amphioctopus fangsiao” should be used in italics. And, “Ca2+”, “5×103” should be used in superscripts in the manuscript. “EC50”, “IC50” and “CO2” should be used in subscripts in the manuscript. “℃” should be used in its current form.

à We corrected Conus tulipa, Lasius niger, Octopus vulgaris, O. ocellatus, Amphioctopus fangsiao as italicized in main text. Ca2+, 5×103 corrected to superscripts in main text. EC50, IC50 and CO2 corrected to subscripts in main text.

  1. “in vitro” and “in vivo” should be used in italics in the manuscript.

à We corrected in vitro and in vivo as italicized.

  1. Comparisons with similar previous studies should be made in the discussion section, such as EC50, IC50, and so on.

à We thank the reviewer for guiding us to improve the discussion. In the revised manuscript, we added comparisons with similar previous studies as follows: “For reference, EC50 of vasopressin on human V1bR and V2R is 1.51 nM and 2.87 nM and that of desmopressin on human V1bR and V2R is 11.4 nM and 23.9 nM, respectively[8]. EC50 of CPT was 10.02 nM on human V1bR and 12.09 nM on human V2R, indicating similar potency with desmopressin (Figure 2). EC50 of vasopressin is 1.31 nM on rat V1bR and 43.5 nM on rat V2R[8].”

Round 2

Reviewer 2 Report

Although the authors have corrected some conclusions with more cautious attitude while I still insist on that necessary experiments about stability comparison of different peptides activating V1bR and V2R to further proving the value of this research. In addition, the present conclusion about what CPT activates V1bR and V2R is derived from an indirect evidence, thus I think it necessary to provide more evidences it. For example, provide evidences that CPT interacts with V1bR and V2R like itc, SPR and so on.

Author Response

à We thank the reviewer for valuable comments and suggestions to enhance the value of this research.

  • Regarding the stability comparison of different peptides, we are sorry that we cannot show experimental results. Of course, stability is usually an important issue in peptide research. We have plans for pharmacokinetic profiling as a part of the next project, preclinical development. Instead, this time we speculated the stability by in silico analysis of peptide properties. Online peptide analyzing tools provided by Thermo Fisher Scientific Inc. and GenScript Inc. were used for this work. As results, the natural peptides (OXT, AVP, OTP, CPT) used in this study did not have remarkable differences in hydrophobicity, molecular weight, and isoelectric point among them. In addition, there was no significant difference in the net charge at pH7 or content of less stable amino acids. When administered intravenously in vivo, the half-lives of OXT and AVP are generally known to be several minutes. In case of desmopressin, deaminated of N-terminus and a presence of D-form amino acid (D-Arg8) greatly extend its half-life to several hours. CPT has the same chemical structure as AVP except for only 2 out of 9 amino acids. We did not find any clues that OTP or CPT would have significantly different half-life compared to OXT or AVP. In the revised manuscript, we corrected the previous modifications to be more detail by adding above explanations in 3rd paragraph of the Discussion part as follows:

“When administered intravenously, the half-life of AVP is generally known to be several minutes. In case of desmopressin, deaminated of N-terminus and a presence of D-form amino acid greatly extend its half-life to several hours. We could not find any clues that CPT would have significantly different half-life compared to AVP. CPT has the same chemical structure as AVP except for only 2 out of 9 amino acids. According to in silico analysis of amino acid sequences using online peptide analyzing tools (provided by Thermo Fisher Scientific Inc. and GenScript Inc.), there is no remarkable difference between CPT and AVP in terms of hydrophobicity, molecular weight, isoelectric point, net charge at pH7, and the content of less stable amino acids. Theses suggests CPT will have a similar stability and a natural, short half-life comparable to that of AVP.”

  • With regard to providing more evidence on how CPT activates V1bR and V2R, we would like to ask for the reviewer’s understanding that we cannot present binding affinity data such as ITC (Isothermal Calorimetry) and SPR (Surface Plasmon Resonance) at this time. Referring to the previous studies cited in the discussion, other natural analogs of OXT/AVP, conopressin-T and inotocin, showed receptor binding experimentally. Likewise, the major acting mechanism of OTP and CPT, which are also natural analogs of OXT/AVP and similar to conopressin-T or inotocin, can be considered as binding to the receptors. The plan of our project was to use rat to integrate the results with human receptors. This project was over, and in the next project there are plans to perform toxicity test and detailed mechanism study. More detail, our next plan includes EC50 and dissociation constant (Ki) on rat V1bR and V2R for evaluating current efficacy and providing dose finding range for acute and repetitive toxicity. We will be grateful if the reviewer understands our next plan.

Reviewer 3 Report

It can be accepted in the current form.

Author Response

3rd reviewer

It can be accepted in the current form.

à We appreciate your acceptance.

Round 3

Reviewer 2 Report

i have no furhter comments.